# SUPERPIPELINE: A UNIVERSAL APPROACH FOR REDUCING GPU MEMORY USAGE IN LARGE MODELS

## ABSTRACT

The rapid growth in size and complexity of machine learning models, particularly in natural language processing and computer vision, has led to significant challenges in model execution on hardware with limited resources. This paper introduces Superpipeline, a novel framework designed to optimize the execution of large-scale AI models on constrained hardware for both training and inference phases. Our approach focuses on dynamically managing model execution by partitioning models into individual layers and efficiently transferring these partitions between GPU and CPU memory. Superpipeline achieves substantial reductions in GPU memory consumption—up to 60% in our experiments—while maintaining model accuracy and acceptable processing speeds. This enables the execution of models that would otherwise exceed available GPU memory capacity. Unlike existing solutions that primarily target inference or specific model types, Superpipeline demonstrates broad applicability across large language models (LLMs), vision-language models (VLMs), and vision-based models. We evaluate Superpipeline's effectiveness through comprehensive experiments on diverse models and hardware configurations. Our method is characterized by two key parameters that allow fine-tuning of the trade-off between GPU memory usage and processing speed. Importantly, Superpipeline does not require model retraining or parameter modification, ensuring full preservation of the original model's output fidelity. The simplicity and flexibility of Superpipeline make it a valuable tool for researchers and practitioners working with state-of-the-art AI models under hardware constraints. It enables the use of larger models or increased batch sizes on existing hardware, potentially accelerating innovation across various machine learning applications. This work represents a significant step towards democratizing access to advanced AI models and optimizing their deployment in resource-constrained environments.

## 1 INTRODUCTION

The field of machine learning has undergone unprecedented growth in recent years, with neural network models at the forefront of this revolution. These models, spanning domains from natural language processing to computer vision, have demonstrated remarkable capabilities in tackling complex tasks. However, their increasing size and complexity present significant challenges for execution, particularly in resource-constrained environments. State-of-the-art models such as LLaMA-3 Dubey et al. (2024) and PaLM 2 Anil et al. (2023) now comprise hundreds of billions of parameters, pushing the boundaries of what's possible in language understanding and generation. While these models achieve unprecedented performance across a wide range of tasks, they also demand substantial computational resources, straining the limits of current hardware capabilities. As model parameters reach into the hundreds of billions, the constraints of GPU memory become a critical bottleneck, especially during both training and inference tasks on consumer-grade hardware. This growing disparity between model size and available computational resources presents a pressing challenge for the machine learning community, necessitating innovative solutions for efficient model execution, training, and deployment.

The machine learning community has made significant strides in optimizing model training on high-performance computing clusters. Techniques such as model parallelism Shoeybi et al. (2019), which distributes model layers across multiple devices, and data parallelism, which processes different

batches of data on separate devices, have been crucial in scaling up model sizes. Recent advancements like Fully Sharded Data Parallel (FSDP) Zhao et al. (2023) and Distributed Data Parallel (DDP) Li et al. (2020) have further improved training efficiency by optimizing memory usage and communication patterns. FSDP, in particular, allows for training larger models by sharding parameters, gradients, and optimizer states across data parallel workers. However, while these techniques have revolutionized training capabilities, they primarily address the needs of institutions with access to substantial computational resources. For the broader user base, both training and inference — the process of deploying trained models to make predictions on new data — have become increasingly challenging, especially when such models must run on consumer-grade hardware or edge devices with constrained computational resources. Even when high-end hardware is available, AI practitioners often run into out of memory (OOM) issues when dealing with large batch sizes that are critical for producing high-performance models.

Recent advances in model optimization have addressed the challenges of working with large models, focusing on both efficient training and inference. Model segmentation and partitioning techniques, such as GPipe Huang et al. (2019) and Megatron-LM Shoeybi et al. (2019), enable the distribution of large models across multiple accelerators. Dynamic memory management strategies, like the Zero Redundancy Optimizer (ZeRO) Rajbhandari et al. (2020) and SuperNeurons Wang et al. (2018), optimize memory usage during training by minimizing data redundancy and efficiently managing intermediate activations. Pipelined execution methods such as PipeDream Narayanan et al. (2019) and TeraPipe Li et al. (2021) have shown considerable promise in improving throughput for distributed training. In the realm of inference, recent work has made significant strides in addressing efficiency challenges. Alizadeh et al. Alizadeh et al. (2023) propose an innovative method to run LLMs on devices with limited DRAM capacity by utilizing flash memory for model storage. The FlexGen system by Sheng et al. Sheng et al. (2023) addresses the challenge of running LLMs on a single commodity GPU with limited memory by utilizing a combination of GPU, CPU, and disk storage. While these advancements represent significant progress, many existing techniques are specifically tailored for LLMs and may not generalize well to other types of neural network architectures. Additionally, some approaches may produce outputs that differ from the original model, potentially affecting performance and reliability.

In this paper, we present Superpipeline, a novel approach designed to overcome the limitations associated with executing and training large neural network models on limited hardware resources. Our method synthesizes and extends existing concepts to formulate a comprehensive framework that addresses both memory constraints and execution efficiency, while maintaining three crucial advantages. First, our approach ensures perfect fidelity to the original model's output, guaranteeing that the results of both training and inference are identical to those produced by the unmodified model. Second, our method is designed for versatility, easily adaptable to a wide range of neural network architectures beyond just LLMs. This broad applicability makes our solution relevant across various domains and model types. Third, we prioritize ease of use, allowing for straightforward implementation without the need for complex model modifications or specialized hardware setups. Referring to Figure 1, Superpipeline is reminiscent of the super pipelining technique in computer architecture Gaudiot et al. (2005). Superpipeline breaks a model into units and load $k$ units into GPU memory initially. Once a preset number, $k'$, where $k' < k$, of units have been executed in GPU, they are offloaded back to CPU to make space for the next $k'$ units while $k - k'$ units are still executing in GPU. The key contributions of Superpipeline can be summarized as follows:

1. **Efficient Training and Inference:** Our method enhances both training and inference phases, ensuring optimized execution on single GPU environments. It addresses the critical need for efficiently training and deploying large models in resource-constrained scenarios.

2. **No Model Retraining or Parameter Modification:** Our method works without introducing any new parameters to the model, ensuring that no retraining is required. This guarantees that both the model structure and its output remain identical to the original.

3. **Universal Applicability:** We present a versatile approach that is easily adaptable to various neural network architectures, from LLMs to image generation models like Stable Diffusion, without requiring model-specific modifications.

4. **Simplified Implementation and Broad GPU Compatibility:** Our method is designed for straightforward implementation, requiring no complex modifications or specialized hardware setups. Additionally, unlike methods such as FlashAttention Dao et al. (2022), which

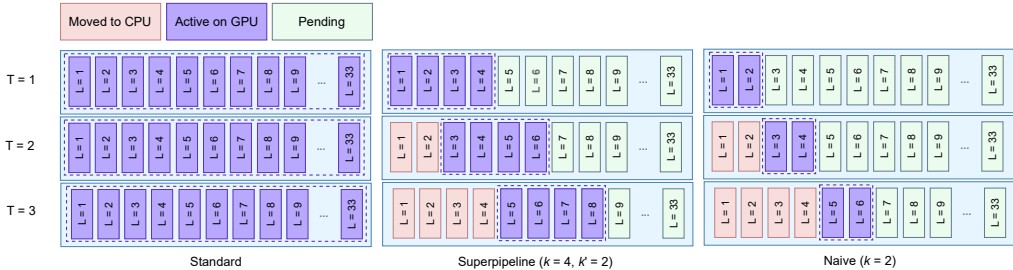

Figure 1: Superpipeline Diagram. Comparison of model execution strategies: Standard (all layers on GPU), Naive ($k = 2$), and Superpipeline ($k = 4, k' = 2$). $k$ represents layers simultaneously on GPU. $k'$ denotes layers transferred back to CPU after computation, and simultaneously, the number of next layers moved to GPU. Superpipeline optimizes GPU memory usage through this dynamic layer management.

> are limited to Ampere GPUs, our approach is compatible with any GPU architecture, providing greater flexibility and accessibility across various hardware setups.

By focusing on these key aspects, Superpipeline offers a practical and efficient solution for training and deploying large models on memory-constrained devices, effectively balancing computational load and memory availability to maximize performance without sacrificing accuracy or generalizability. This has profound implications for various applications, including edge computing, mobile applications, large batch-sized training recipes, and other scenarios where access to high-end computing resources is limited. By enabling the training and deployment of advanced neural networks on such devices, our method can help bridge the gap between cutting-edge AI research and practical, everyday applications as well as ensuring equity among common AI practitioners and well-endowed institutions alike.

The remainder of this paper is organized as follows: Section 2 provides a comprehensive review of related work in model optimization and efficient training and inference techniques. Section 3 details our proposed method, emphasizing its universality and output fidelity preservation for both training and inference phases. Section 4 presents our experimental results across various model types and tasks, demonstrating the effectiveness of Superpipeline in both training and inference scenarios. We conclude in Section 6 with a summary of our findings and their potential impact on democratizing access to state-of-the-art AI models.

## 2 RELATED WORK

Recent advancements in neural network research have focused on enhancing the efficiency and scalability of large models, particularly in environments with limited hardware resources. This section reviews key developments in model compression, memory management, parallelism strategies, and data transfer optimization techniques relevant to our proposed method.

### 2.1 MODEL SEGMENTATION AND PARTITIONING

The concept of dividing large models into smaller, manageable units has gained prominence in recent years. GPipe Huang et al. (2019) introduced a scalable model-parallelism library that efficiently trains large neural networks using pipeline parallelism. By partitioning deep networks into smaller segments and distributing them across different accelerators, GPipe optimizes hardware utilization and reduces training time. To maintain the simplicity of the proposed method and ensure its generalizability across different models, we use the repetitive layers present in every deep model as the model's partition for memory management.

Megatron-LM Shoeybi et al. (2019) proposed an intra-layer model parallelism technique that efficiently trains large-scale Transformer-based language models by distributing computations across multiple GPUs. While this method enhances scalability for training, our approach adapts these prin-

ciples for single-GPU environments, focusing on dynamic partitioning and memory management to optimize inference and training processes.

## 2.2 Model Compression and Selective Execution

As large language models (LLMs) increase in size, reducing their computational and memory requirements has become a critical area of research. Model compression techniques such as pruning and quantization have been extensively explored to shrink models without significantly compromising performance Han et al. (2015); Jaiswal et al. (2023); Ahmadian et al. (2023); Li et al. (2024). Additionally, selective execution methods, including sparse activations and conditional computation Zhang et al. (2024); Baykal et al. (2024), aim to reduce the computational overhead by limiting operations to necessary components, which aligns with the broader goal of minimizing resource usage during inference.

Selective weight loading is another related concept, where techniques have been developed to dynamically load a subset of weights based on activation patterns Liu et al. (2023); Sheng et al. (2023). This strategy reduces the memory footprint required for model execution, complementing efforts to manage memory transfers between different hardware components effectively.

## 2.3 Dynamic Memory Management and Hardware Optimization

Dynamic memory management strategies have been proposed to address GPU memory limitations in training and deploying deep neural networks. The Zero Redundancy Optimizer (ZeRO) Rajbhandari et al. (2020) optimizes memory usage by eliminating redundant copies of model states and distributing them across devices. This method has parallels to dynamic memory management strategies used to optimize memory allocation for inference, particularly in settings with limited hardware resources.

Hardware optimization techniques, including efficient memory architectures Gao et al. (2019) and dataflow optimizations Han et al. (2016), also contribute to more efficient LLM inference. These methods can further enhance algorithmic improvements for memory management and model execution by leveraging hardware-specific optimizations.

## 2.4 Pipelined Execution and Speculative Techniques

Pipelined execution has been a focus of several studies aimed at improving deep neural network (DNN) training throughput. PipeDream Narayanan et al. (2019) and TeraPipe Li et al. (2021) explore combining intra-batch and inter-batch parallelism to optimize training processes across multiple GPUs. In contrast, adaptations of pipelining principles for single-GPU environments have also been proposed to enhance inference efficiency, where models are partitioned and dynamically transferred between memory hierarchies to optimize execution speed.

Speculative execution, a technique used to manage latency in model inference, has been explored in various contexts, including speculative decoding for LLMs Zhang et al. (2023); He et al. (2023). This approach utilizes draft models to predict outputs and verifies them with larger models, serving as an orthogonal strategy to improve inference efficiency. Speculative techniques and adaptive execution methods contribute to the growing toolbox for managing the complexity of large models on constrained hardware.

## 2.5 Transfer Strategies and Pipeline Optimization

Optimizing data transfer between different memory hierarchies is a critical yet underexplored area for efficient large model inference. Research on minimizing memory usage through optimal checkpointing and data movement Feng & Huang (2021) provides a foundation for strategies that aim to reduce data transfer overhead during model execution. Techniques that streamline these transfers are essential for executing large models effectively, particularly on devices with limited GPU or DRAM capacity.

In contrast to previous works, which primarily target specific model types like LLMs or focus on optimizing either the training or inference phase, Superpipeline is versatile and applicable across

a wide range of models. It seamlessly integrates into both the training and inference processes without altering the original model's output, making it a simple yet effective solution for enhancing efficiency on resource-constrained hardware. Additionally, it provides AI researchers with an efficient solution for developing their models on high-end GPUs, enabling the use of larger batch sizes while optimizing resource utilization.

## 3 PROPOSED METHOD: SUPERPIPELINE

This section introduces Superpipeline, our novel approach for efficient execution of large neural network models on constrained hardware resources. Superpipeline addresses the challenge of running memory-intensive models on limited GPU hardware through dynamic memory management and optimized data transfer strategies.

### 3.1 CONCEPTUAL FRAMEWORK

Our method segments large models into manageable units based on their repetitive structure. This approach, applicable to various neural network architectures, enables efficient processing and dynamic memory management. By exploiting the inherent repetition in modern models, we achieve simplicity in implementation and universality across model types.

### 3.2 KEY COMPONENTS OF SUPERPIPELINE

#### 3.2.1 MODEL SEGMENTATION STRATEGY

We partition neural networks along their natural repetitive boundaries, such as transformer layers in language models or convolutional blocks in vision models. Each repetitive unit becomes a distinct partition. This strategy requires minimal modification to the original architecture, adapts to different model sizes, and preserves model behavior. For example, a model like LLaMA-2 7B with 32 repeating layers would yield 32 partitions. This approach forms the foundation for our subsequent optimization techniques, allowing efficient resource management across diverse model types.

#### 3.2.2 DYNAMIC GPU-CPU PARTITION TRANSFER

Superpipeline employs a dynamic approach to memory management. Only specific partitions are loaded onto the GPU as needed, and once computation is complete, their outputs are transferred back to CPU memory. This process frees up GPU memory for subsequent partitions, allowing for the processing of models that would otherwise exceed available hardware capacity. This dynamic transfer mechanism is crucial for optimizing GPU resource utilization. It allows larger models to be run on more constrained hardware by effectively managing the limited GPU memory available.

### 3.3 THE SUPERPIPELINE ALGORITHM

Superpipeline introduces two critical hyperparameters: $k$, representing the number of partitions simultaneously on the GPU, and $k'$, which denotes the number of partitions transferred back to the CPU after computation, making room for the next $k'$ partitions. Figure 1 illustrates this.

By adjusting these parameters, the Superpipeline framework achieves an optimal balance between GPU memory usage and processing speed. Increasing $k$ maximizes GPU utilization and accelerates computation, but also raises memory requirements. On the other hand, decreasing $k$ lowers memory usage while slowing down execution. This flexibility allows the method to be tailored to specific hardware constraints, optimizing the trade-off between speed and memory efficiency.

In the training phase, Superpipeline extends its benefits to both the forward and backward passes. During the forward pass, it dynamically transfers partitions between GPU and CPU as needed. The same process is repeated for the backward pass, where gradients are computed. This dual application in both forward and backward passes results in even greater reductions in GPU memory usage while maintaining acceptable performance.

By efficiently managing memory across both phases of training, Superpipeline significantly reduces the overall GPU memory footprint, particularly in large-scale models. This method ensures that even

resource-constrained hardware can support models that would otherwise be unmanageable, without sacrificing speed or accuracy.

# 4 EXPERIMENTS AND RESULTS

In this section, we present our experimental methodology and key findings. Our experimental setup encompasses a diverse array of models, ranging from vision architectures to language models, all implemented using Superpipeline. This broad selection demonstrates the versatility and wide applicability of our proposed method. We begin by outlining the implementation details and experimental parameters, followed by a comprehensive description of the models tested. Through these experiments, we aim to demonstrate two critical aspects of our approach: first, its ability to reduce GPU memory usage significantly, and second, its capacity to maintain acceptable inference times across various model types. Our experiments are designed to illustrate not only the ease with which our approach can be adapted to various model architectures and domains but also its effectiveness in optimizing resource utilization without substantially compromising performance.

## 4.1 EXPERIMENTAL SETUP

**Models.** To demonstrate that the method presented in this work is applicable to any model, we have conducted our evaluation across different categories of models. We have selected three different models from three distinct domains. One is the llama2 model from the world of LLM (Large Language Models), the SD model from the world of VLM (Vision Language Models), and ViT-bigG from the world of vision models. We perform our evaluations of the proposed method in two sections: during inference time and during training time. The aim of these experiments is to show the extent to which the proposed method helps in optimizing GPU consumption and how much faster it is compared to the naive approach.

**Hardware Configuration.** We evaluated models on three distinct hardware configurations to ensure the generalizability of our method across various devices. The first setup featured a Quadro 8000 graphics card with 50 GB of GPU memory. The second configuration utilized an NVIDIA GTX 3090 graphics card, offering 24 GB of GPU RAM. Our third setup employed an H20 graphics card with a substantial 98 GB of GPU RAM. By conducting evaluations across these diverse hardware environments, we aimed to validate the robustness and adaptability of our approach

## 4.2 RESULTS

Our experiments evaluated Superpipeline across four distinct modes of operation:

1. **Standard mode:** The entire model is loaded onto the GPU and processed, representing the conventional approach for model execution.

2. **Naive method:** The model is loaded onto the GPU $k$ layers at a time, offering a simple but potentially inefficient way to reduce memory usage.

3. **CPU-only mode:** The entire model runs on the CPU without GPU acceleration, providing a baseline for comparison in resource-constrained environments.

4. **Superpipeline method:** Our proposed approach for dynamic memory management, balancing GPU utilization and processing efficiency.

The key metrics we focused on were GPU Memory Usage and Processing Time. GPU Memory Usage, measured in gigabytes (GB), shows how efficiently each method utilizes available GPU memory. Processing Time, measured in milliseconds (ms) for inference tasks and iterations per second for training tasks, reflects the speed of each method. It's important to note that Superpipeline, by design, does not alter the model's computations or outputs in any way. The results produced by Superpipeline are *identical to those of the standard mode*, ensuring perfect fidelity to the original model's performance and accuracy.

Table 1: Superpipeline Performance During Inference

| Model | Method | GPU Usage (GB) | Time (ms) | K | K' |
|---|---|---|---|---|---|
| ViT-bigG | Standard | 13.7 | 37.5 ms /embed | - | - |
| | CPU-only | 0 | 9250 ms /embed | - | - |
| | Naive | 4.2 | 181.5 ms /embed | 1 | - |
| | Naive | 5.2 | 175.5 ms /embed | 8 | - |
| | Naive | 6.0 | 176.6 ms /embed | 15 | - |
| | Superpipeline | 4.8 | 111.5 ms /embed | 4 | 2 |
| | Superpipeline | 5.7 | 109.7 ms /embed | 6 | 3 |
| | Superpipeline | 6.0 | 103.5 ms /embed | 10 | 8 |
| | Superpipeline | 7.3 | 96.8 ms /embed | 14 | 12 |
| | Superpipeline | 8.8 | 90.5 ms /embed | 19 | 16 |
| | Superpipeline | 10.7 | 72.5 ms /embed | 19 | 16 |
| LlaMA2 | Standard | 15.0 | 26 ms /token | - | - |
| | CPU-only | 0 | 29200 ms /token | - | - |
| | Naive | 2.9 | 4520 ms /token | 1 | - |
| | Naive | 3.8 | 4000 ms /token | 8 | - |
| | Naive | 3.8 | 3980 ms /token | 8 | - |
| | Naive | 6.5 | 3970 ms /token | 15 | - |
| | Superpipeline | 4.7 | 2053 ms /token | 4 | 2 |
| | Superpipeline | 5.7 | 1964 ms /token | 5 | 3 |
| | Superpipeline | 8.0 | 1748 ms /token | 8 | 3 |
| | Superpipeline | 9.2 | 1607 ms /token | 10 | 2 |
| | Superpipeline | 11.9 | 1460 ms /token | 10 | 2 |
| | Superpipeline | 13.0 | 880 ms /token | 20 | 6 |
| Stable Diffusion | Standard | 6.3 | 10 s /image | - | - |
| | CPU-only | 2.5 | 529 s /image | - | - |
| | Naive | 2.5 | 60 s /image | 1 | - |
| | Naive | 3.8 | 59 s /image | 8 | - |
| | Superpipeline | 2.9 | 33 s /image | 5 | 3 |
| | Superpipeline | 3.3 | 27 s /image | 5 | 4 |
| | Superpipeline | 4.0 | 27 s /image | 8 | 6 |
| | Superpipeline | 4.1 | 22 s /image | 7 | 5 |
| | Superpipeline | 4.4 | 19 s /image | 8 | 2 |
| | Superpipeline | 4.8 | 16 s /image | 9 | 3 |
| | Superpipeline | 5.0 | 14 s /image | 10 | 2 |

### 4.2.1 INFERENCE TIME

One of the significant advantages of our proposed method is its ease of implementation across various existing models by making necessary changes in the forward pass. Since no parameters are added to or removed from the model, and no changes are made to the overall model structure, Superpipeline can be applied to many current models without the need for retraining.

The primary parameters in this approach are $K$ and $k'$. These values can be easily optimized through a grid search, tailored to the hardware on which the model is running. This flexibility allows for adjusting GPU consumption during inference by simply modifying $k$ and $k'$. Consequently, any remaining GPU space can be utilized for processing larger batch sizes if required.

The effectiveness of Superpipeline during inference is demonstrated in Table 1. These results highlight the method's capability to optimize GPU usage without compromising model performance, making it a versatile solution for both training and inference stages.

As shown in Table 1, Superpipeline achieves significant reductions in GPU usage and inference time while maintaining the same accuracy as the standard and naive methods. This demonstrates the

method's efficiency in resource utilization during the inference phase. Table 4 shows results from the Quadro GPU, with other GPU results in the appendix.

The adaptability of Superpipeline to different hardware configurations and model architectures, combined with its performance benefits in both training and inference, positions it as a valuable tool for optimizing deep learning workflows across various applications and deployment scenarios.

### 4.2.2 TRAINING TIME

Unlike some previous methods that are only applicable during the inference stage, Superpipeline can be used in both training and inference phases with minimal modifications.

Implementing Superpipeline involves applying this method to the `forward` section of each model. By utilizing `pre_backward_hook` and `post_backward_hook` functions, Superpipeline can be easily integrated into the model training phase. This capability is particularly significant during training, as gradients are calculated in addition to the usual computations. In these conditions, the efficiency of our proposed method in optimizing GPU usage becomes even more pronounced.

A key feature of Superpipeline is the preservation of model accuracy even when used in training. Since no changes are made to the computational values, the model's output using our proposed method is identical to that of the standard approach. This distinguishes Superpipeline from methods that rely on predicting which neurons or layers will be used, which may lead to prediction errors and changes in output.

To rigorously evaluate the proposed method, we compared the ViT-BigG model on the imagenet-tiny dataset using identical hyperparameters (batch size, learning rate, number of epochs). The

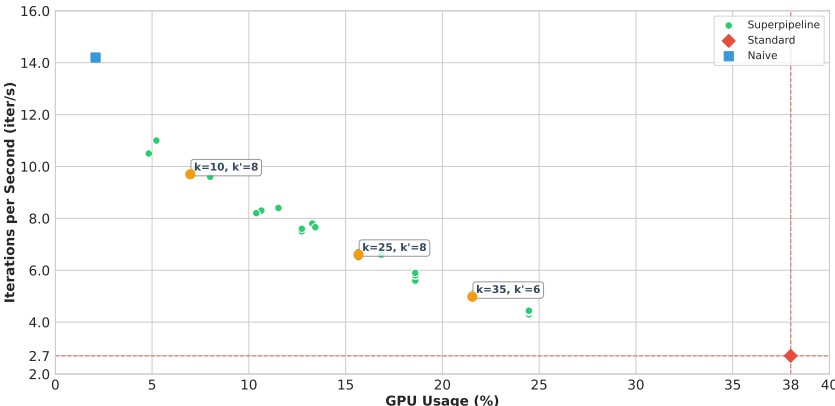

Figure 2: Comparison of memory usage and speed during ViT-BigG training on ImageNet-tiny.

results of this comparison are shown in Figure 2. As observed, the Superpipeline method not only significantly reduces GPU consumption but also provides a highly acceptable speed compared to the standard mode and the naive method.

Superpipeline offers several notable advantages. It's implementation process for training remains straightforward, and can be applied to various types of models. Additionally, by adjusting the parameters $k$ and $k'$, GPU consumption can be easily controlled. By optimizing GPU usage, it becomes possible to train models with larger batch sizes, which can lead to improved performance and faster model convergence.

### 4.2.3 BENEFIT OF DIFFERENT GPU USAGE

While Superpipeline offers significant benefits for general users, its impact on AI research and model development is particularly noteworthy. As deep learning models continue to grow in size and complexity, GPU memory constraints have become a critical bottleneck in the training process. Even with high-capacity GPUs boasting 50 to 100 gigabytes of memory, researchers face limitations in increasing batch sizes, a crucial factor for many advanced training techniques.

Table 2: GPU Usage for ViT-BigG and LLaMA2 Models w/ and w/o Gradient Checkpointing. OOM: Out-Of-Memory.

| Model | Method | With Grad. Checkpointing | | | | Without Grad. Checkpointing | | | |
|---|---|---|---|---|---|---|---|---|---|
| | | BS | GPU | BS | GPU | BS | GPU | BS | GPU |
| **ViT-BigG** **(Fully Trainable)** (on Quadro) | Superpipe ($k$=6, $k'$=3) | 16 | 10.1 | 64 | 25.8 | 4 | 23.0 | 10 | 42.3 |
| | | 32 | 15.4 | 128 | 42 | 8 | 36.9 | 12 | 48.0 |
| | Standard | 16 | 38.9 | 64 | 47.4 | 4 | 42.6 | 10 | OOM |
| | | 32 | 41.8 | 128 | OOM | 8 | OOM | 12 | OOM |
| **LLaMA2** **(Fully Trainable)** (on H20) | Superpipe ($k$=6, $k'$=3) | 32 | 33.5 | 128 | 53.6 | 16 | 55.3 | 64 | OOM |
| | | 64 | 39.5 | 256 | 81.8 | 32 | 85.6 | 128 | OOM |
| | Standard | 32 | OOM | 128 | OOM | 16 | OOM | 64 | OOM |
| | | 64 | OOM | 256 | OOM | 32 | OOM | 128 | OOM |
| **LLaMA2** **(Half of Layers Frozen)** (on H20) | Superpipe ($k$=6, $k'$=3) | 4k | 21 | 16k | 48 | 2k | 29.8 | 8k | 73 |
| | | 8k | 30 | 32k | 88.8 | 4k | 44 | 10k | 88.3 |
| | Standard | 4k | 36 | 16k | 64 | 2k | 64 | 8k | OOM |
| | | 8k | 45 | 32k | OOM | 4k | 79 | 10k | OOM |

As shown in Table 2, Superpipeline significantly expands the potential for larger batch sizes during model training. For instance, when training the LLaMA2 model without gradient checkpointing, the standard approach fails due to out-of-memory errors even at smaller batch sizes. In contrast, Superpipeline successfully trains the model with larger batch sizes, demonstrating its ability to handle scenarios infeasible with standard training methods.

To provide a more equitable comparison and further demonstrate Superpipeline's advantages in enabling larger batch sizes, we conducted an additional experiment where half of the LLaMA2 model's layers were frozen. This approach allowed the standard method to handle larger batch sizes, creating a more balanced comparison scenario. In this setting, Superpipeline continued to outperform, accommodating significantly larger batch sizes and achieving more efficient GPU utilization.

By alleviating memory constraints, Superpipeline enables the exploration of training regimes that were previously infeasible, potentially accelerating advancements in areas such as self-supervised learning, large-scale visual representation learning, and the training of Large Language Models. This adaptability is crucial in an era where model innovation often outpaces hardware advancement, allowing researchers with limited resources to work on cutting-edge models and training techniques previously exclusive to well-resourced institutions.

## 5 LIMITATIONS AND FUTURE WORK

An examination of Table 1 reveals that while the superpipeline method consistently outperforms the naive approach across all models, the performance gap between the proposed superpipeline method and the standard approach is notably smaller for the ViT-bigG model compared to models like Llama2. To investigate this discrepancy, we measured two distinct timings for both the ViT-bigG and Llama2 models: the time required to transfer a layer to the GPU and the time needed to transfer a layer to the CPU. The results are illustrated in Figure 3. Two key observations can be drawn from Figure 3. First, the transfer time of a layer to the CPU is slower than to the GPU. Second, and more importantly, we observe that the transfer speed of a single layer from the Llama model is significantly slower than that of the ViT-bigG model. This difference explains the larger performance gap between the superpipeline and standard approaches in the Llama model.

In essence, when considering a single forward pass, the superpipeline and standard methods do not differ significantly. However, since we calculate model speed based on an average of multiple consecutive forward passes, a limitation becomes apparent in the superpipeline approach. Although the model's output is quickly generated in the first forward pass, it cannot immediately produce the second output as it must wait for the layers from the previous forward pass to complete their transfer to the CPU. This issue represents a key limitation of our work. In scenarios where the layer transfer speed to the CPU is slow for a particular model or hardware configuration, the superpipeline method, while still outperforming the naive approach, may not achieve performance parity with the standard method.

Several potential solutions to address this limitation could be explored in future work. One approach involves rewriting the model transfer function to the CPU using CUDA custom kernels. Another possibility is developing a faster method for creating and transferring a copy of each layer to the GPU. This approach would eliminate the need to transfer layers back to the CPU after GPU processing, instead overwriting the previous layer directly on the GPU. Currently, implementing this with existing PyTorch features is significantly more time-consuming than transferring a layer to the CPU, necessitating a more optimized implementation. In future research, we plan to explore these optimization strategies to further enhance the performance of the superpipeline method across a wider range of models and hardware configurations. Additionally, we aim to investigate the applicability of our approach to emerging model architectures and to develop adaptive strategies that can automatically adjust the superpipeline parameters based on the specific characteristics of the model and hardware in use.

## 6 Conclusion

In this paper, we introduced Superpipeline, a novel method for efficient execution of large neural network models on constrained hardware resources. Our approach addresses the critical challenge of deploying and training increasingly complex models in environments with limited GPU memory, without compromising model performance or accuracy. The key strengths of Superpipeline lie in its versatility and ease of implementation. Unlike previous methods that primarily focused on LLM models or were limited to inference stages, Superpipeline demonstrates broad applicability across various model architectures, including LLMs, VLMs, and vision-based models. Moreover, it can be seamlessly integrated into both inference and training pipelines, offering a comprehensive solution for resource optimization throughout the model lifecycle. A significant advantage of our method is its ability to substantially reduce GPU memory consumption while maintaining acceptable execution speeds. This is

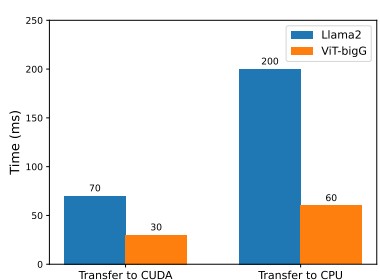

Figure 3: Comparison of layer transfer times between GPU and CPU for ViT-bigG and Llama2 models.

achieved without adding new parameters to the model or requiring retraining, ensuring that the model's output in Superpipeline mode remains identical to that in standard mode. This preservation of accuracy sets Superpipeline apart from other optimization techniques that may introduce performance trade-offs.

Our experimental results across diverse model types and hardware configurations validate the effectiveness of Superpipeline. We demonstrated significant reductions in GPU usage during both inference and training, with minimal impact on processing speed. The method's adaptability to different hardware setups further enhances its practical value, making it a viable solution for a wide range of deployment scenarios. The simplicity of Superpipeline's implementation, coupled with its flexibility in fine-tuning through the k and k' parameters, positions it as a powerful tool for researchers and practitioners alike. By optimizing resource utilization, our method opens up new possibilities for working with larger models or increased batch sizes on existing hardware, potentially accelerating research and development in the field of deep learning.

In conclusion, Superpipeline represents a significant step forward in making advanced AI models more accessible and efficient to deploy. As the complexity of neural networks continues to grow, methods like Superpipeline will play a crucial role in bridging the gap between state-of-the-art model architectures and the practical constraints of real-world computing environments. Future work could explore further optimizations and extensions of this approach, potentially leading to even more efficient and scalable solutions for large-scale model deployment and training. You may include other additional sections here.

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

# A APPENDIX

## A.1 RESULTS ON DIFFERENT GPUS

In this section, we present the results of applying the Superpipeline method on two different GPUs: the NVIDIA RTX 3090 and the H20 (shown in Table 3. The Superpipeline approach was evaluated using three different models—ViT-bigG, LlaMA2, and Stable Diffusion—under varying memory constraints and batch sizes

## A.2 OPTIMIZED PARTITION TRANSFER STRATEGY

Our experiments revealed that the method of transferring partitions between GPU and CPU significantly impacts overall performance. We compared two approaches: Sequential Transfer and Batch Transfer. In Sequential Transfer, layers are transferred one-by-one to the GPU and back to the CPU. Batch Transfer, on the other hand, moves all layers to the GPU simultaneously, then back to the CPU as a batch. As illustrated in Figure 4, the batch transfer method proved significantly faster, despite

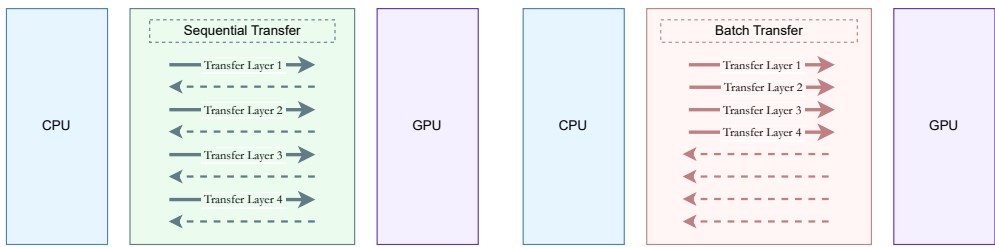

Figure 4: Comparison of Sequential and Batch Transfer Strategies

involving the same number of total transfers. Figure 5 provides empirical evidence of this performance difference across various model architectures. These findings underscore the importance of

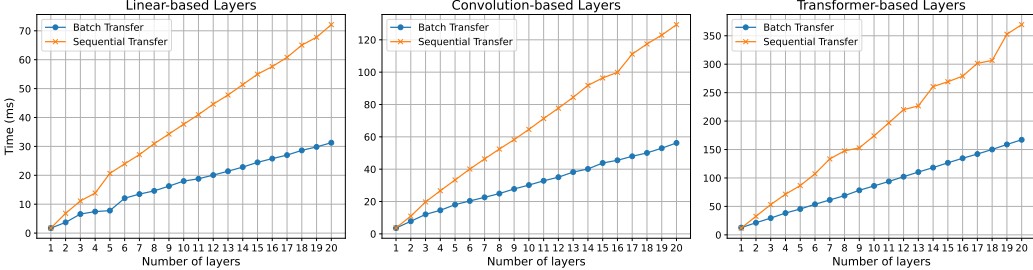

Figure 5: Performance comparison of Sequential vs. Batch Transfer strategies

optimizing not just the partitioning of the model, but also the mechanisms for data transfer between different memory hierarchies.

Table 3: Superpipeline Performance During Inference On RTX 3090

| Model | Method | GPU Usage (GB) | Time (ms) | K | K' |
|---|---|---|---|---|---|
| | Superpipeline | 4.1 | 242.8 ms /embed | 4 | 3 |
| | Superpipeline | 5.7 | 223.25 ms /embed | 5 | 3 |
| ViT-bigG | Superpipeline | 8 | 212.5 ms /embed | 7 | 5 |
| | Superpipeline | 8.8 | 213.1 ms /embed | 9 | 4 |
| | Superpipeline | 11.9 | 197.5 ms /embed | 11 | 7 |
| | Superpipeline | 4.8 | 108.1 ms /token | 4 | 2 |
| | Superpipeline | 5.1 | 104.6 ms /token | 6 | 4 |
| LlaMA2 | Superpipeline | 5.5 | 100.5 ms /token | 7 | 6 |
| | Superpipeline | 6.4 | 98.3 ms /token | 11 | 8 |
| | Superpipeline | 7.7 | 91.2 ms /token | 16 | 12 |
| | Superpipeline | 8.6 | 86.2 ms /token | 18 | 16 |
| | Superpipeline | 2.9 | 70 s /image | 3 | 2 |
| | Superpipeline | 3.8 | 54 s /image | 6 | 2 |
| Stable Diffusion | Superpipeline | 4.1 | 55 s /image | 8 | 5 |
| | Superpipeline | 4.7 | 53 s /image | 9 | 3 |
| | Superpipeline | 5.0 | 49 s /image | 11 | 2 |

Table 4: Superpipeline Performance During Inference On H20

| Model | Method | GPU Usage (GB) | Time (ms) | K | K' |
|---|---|---|---|---|---|
| | Superpipeline | 4.7 | 34.1 ms /embed | 4 | 3 |
| | Superpipeline | 5.1 | 33.2 ms /embed | 6 | 4 |
| ViT-bigG | Superpipeline | 5.9 | 32.3 ms /embed | 9 | 6 |
| | Superpipeline | 7.2 | 30.3 ms /embed | 14 | 12 |
| | Superpipeline | 8.5 | 28.8 ms /embed | 17 | 16 |
| | Superpipeline | 4.5 | 41.25 ms /token | 4 | 2 |
| | Superpipeline | 5.7 | 40.6 ms /token | 5 | 3 |
| LlaMA2 | Superpipeline | 8.0 | 37.3 ms /token | 7 | 5 |
| | Superpipeline | 7.6 | 36 ms /token | 8 | 2 |
| | Superpipeline | 10.3 | 31.6 ms /token | 11 | 3 |
| | Superpipeline | 11.9 | 30.25 ms /token | 12 | 5 |
| | Superpipeline | 3.3 | 11.7 s /image | 3 | 2 |
| | Superpipeline | 3.8 | 9.2 s /image | 6 | 3 |
| Stable Diffusion | Superpipeline | 4.1 | 8.8 s /image | 8 | 5 |
| | Superpipeline | 4.8 | 8.3 s /image | 9 | 3 |
| | Superpipeline | 5.0 | 7.5 s /image | 10 | 4 |

