# OpenReview forum: "Superpipeline: A Universal Approach for Reducing GPU Memory Usage in Large Models"
_ICLR.cc/2025/Conference — ICLR 2025 Conference Withdrawn Submission_

### Official Review · Reviewer_yKKC · 2024-10-25

**Soundness:** 1
**Presentation:** 1
**Contribution:** 2
**Rating:** 3
**Confidence:** 4

**Summary:**

This paper proposed a GPU processing schedule method to enable process larger models on devices with limited memory by pipelining the layers and swapping between CPU and GPU memory.

**Strengths:**

The method proposed in this work is fair and reasonable. The analysis on the related works is comprehensive.

**Weaknesses:**

1. Poor writing. Some example: 1) Wrong citation format in contents, like “Alizadeh et al. Alizadeh et al. (2023)”; 2) “No Model Retraining or Parameter Modification” and “Simplified Implementation and Broad GPU Compatibility” are features, not contributions; 3) The first line of text on page 3 is indented incorrectly.

2. The evaluation on inference is unproper. This work claims the contribution on enabling small device to run large models for both training and inference. However, all the models used in inference part are smaller than the GPU memory. Can you either use larger model size or batch size to show the benefits of the proposed method on inference? Can you clarify whether LLMs or other models can be inferenced in parallel and be accelerated by the proposed method? If not, this paper should only claim the contribution on training.

3. From the training comparison results shown in Figure 2, the proposed method is much slower than standard GPU processing. Can you judge why it is deserved to train ViT-BigG slower to use less GPU memory? Is this method only beneficial to huge LLMs? If so, this paper should only claim the contribution on LLMs instead of “universal”.

4. The comparison of training is not fair. Can you add the training speed results in table 2 for fair comparison? How much slower is the proposed method than standard GPU processing? Based on the results, can you prove that the proposed method can outperform standard GPU training by processing more tokens at the same time on the same device? Can the conclusion be extended to multiple GPU system with distributed training (including pipeline parallelism, data parallelism, and tensor parallelism)?

5. Lack of innovation. To my knowledge, the key idea of swapping between GPU and CPU memory has already been proposed and researched in many works. The innovation and contribution of this work is a little weak to me.

**Questions:**

1. For the evaluation of inference, can you either use larger model size or batch size to show the benefits of the proposed method on inference? Can you clarify whether LLMs or other models can be inferenced in parallel and be accelerated by the proposed method?

2. For the comparison of training, can you add the training speed results in table 2 for fair comparison? How much slower is the proposed method than standard GPU processing? Based on the results, can you prove that the proposed method can outperform standard GPU training by processing more tokens at the same time on the same device? Can the conclusion be extended to multiple GPU system with distributed training (including pipeline parallelism, data parallelism, and tensor parallelism)?

3. Based on Figure 2, can you judge why it is deserved to train ViT-BigG slower to use less GPU memory? Is this method only beneficial to huge LLMs?

---

### Official Review · Reviewer_7AEk · 2024-10-30

**Soundness:** 1
**Presentation:** 2
**Contribution:** 1
**Rating:** 1
**Confidence:** 4

**Summary:**

The paper presents Superpipeline, a framework for optimizing the execution of large-scale AI models on constrained hardware by dynamically managing memory usage through partitioning models into layers and transferring partitions between GPU and CPU memory. The authors claim that their approach reduces GPU memory consumption by up to 60% while preserving model performance without retraining or parameter changes.

**Strengths:**

The paper is easy to follow.

**Weaknesses:**

The paper has several critical shortcomings that limit its contribution:

**Lack of Comparison with State-of-the-Art Solutions:**
The evaluation section only compares Superpipeline to a baseline approach without benchmarking against more advanced or recent solutions. This omission is problematic, given that many existing methods—some of which are cited in the paper (e.g., FlexGen, PipeDream)—are designed to address similar memory challenges. Without such comparisons, it is difficult to assess whether Superpipeline offers any real advantage.

**Use of a Trivial Cost Model:**
The memory management strategy, which involves swapping data between CPU and GPU based on a simple partitioning scheme, is neither new nor innovative. Similar strategies have been implemented in other works cited in the paper (e.g., GPipe, SuperNeurons). The proposed dynamic transfer mechanism offers no significant improvement beyond these established solutions, raising questions about the novelty of the approach.

**Lack of Technical Depth:**
The paper does not provide any substantial innovation in the cost model for data transfer. The approach relies on well-known techniques without presenting any meaningful modifications or optimizations to improve performance.
In summary, the work fails to demonstrate its novelty by omitting meaningful comparisons with existing methods and relying on a simplistic strategy that has been previously explored. These weaknesses undermine the paper’s claim of contributing a new solution to the problem of memory-constrained model execution.

**Questions:**

Why does the paper not compare to any of the existing solutions (that you listed in the paper)?

In addition to the existing solutions in the paper, there are more:

(1) "Capuchin: Tensor-based GPU Memory Management for Deep Learning", ASPLOS'20.

(2) "SwapAdvisor: Pushing Deep Learning Beyond the GPU Memory Limit via Smart Swapping", ASPLOS'20.

(3) "Scaling distributed deep learning workloads beyond the memory capacity with KARMA", SC'20

---

### Official Review · Reviewer_wfgQ · 2024-11-01

**Soundness:** 2
**Presentation:** 2
**Contribution:** 2
**Rating:** 5
**Confidence:** 3

**Summary:**

The paper propose Superpipeline, a pipeline framework aimed at reducing GPU memory usage for large-scale AI models during training and inference. This framework manages memory by partitioning models into chunks in layerwise, only load some of them to GPU and transferring these between GPU and CPU, achieving up to a 60% reduction in GPU memory usage without compromising model accuracy or processing speed. Hyperparameters of Superpipeline to achieve the best performance to achieve best performance across various architectures. The method is lossless and therefore do not affect convergence.

**Strengths:**

1. The explanations of Superpipeline's mechanism, including the k and k’ parameters for partition management, is very clear and easy to understand.
2. The proposed method is easy to deploy and can be adapted to different hardware by properly adjusting the hyperparameters.
3. The method is similar to a pre-fetch of the following layers and is well-motivated.

**Weaknesses:**

1. In models or hardware configurations where this transfer is slower, the performance gain over standard methods may be reduced
2. The tuning of k and k’ parameters add complexity and require initial experimentation. Can this process be automated selected, such as triton's @autotune method (As stated in Section 5)? What's the possible solution for the automated selection algorithm, can it be solved through Linear Programming?

**Questions:**

See Weakness part.

---

### Official Review · Reviewer_EDqa · 2024-11-02

**Soundness:** 1
**Presentation:** 2
**Contribution:** 1
**Rating:** 1
**Confidence:** 5

**Summary:**

This paper presents Superpipeline, a framework to optimize the execution of large AI models on GPUs with limited memory for training and inference. The main idea is to partition models into so-called units, with respect to layer/block boundaries, and build a three-stage pipeline (CPU-GPU transfer, GPU execution, GPU-CPU transfer) for buffering data at the CPU side to reduce GPU memory cost. Authors claim up to 60% reduction on GPU memory consumption and highlight the correctness, adaptability, and tractability of the approach.

**Strengths:**

+ Focus on a key problem for DL training: the limitation on GPU memory
+ Evaluated on LLM and ViT models
+ Easy-to-follow

**Weaknesses:**

+ No new research contribution. Method is quite simple and well-known
+ No design or implementation details are given
+ Poor evaluation without comparison to other works

Thanks for submitting to ICLR. This work attempts to tackle the major problem of AI training/inference, the lack of GPU memory. It did a reasonable work on summarizing SOTA, but the extremely short description on the design (Section 3, less than a page) did not give any insights about the design and implementation details. From the very limited words, it seems the idea is quite simple: model parallelism with a three-stage pipeline to buffer some data at the CPU side. This is already well-known and did not actually solve the problem, as then the CPU-GPU communication becomes the bottleneck, and workload balance becomes another issue. This may also explain why authors did not compare their approach to any SOTA works exploiting model/pipeline parallelism. They mention generally applicability, but they also did not test on non-NVIDIA GPUs.

**Questions:**

What are the technical contributions?
How can you demonstrate pushing forward the filed without comparing to alternative works?

---

### Official Review · Reviewer_1ojT · 2024-11-05

**Soundness:** 3
**Presentation:** 3
**Contribution:** 2
**Rating:** 5
**Confidence:** 3

**Summary:**

The paper introduces Superpipeline, a novel framework aimed at optimizing memory usage in large-scale AI models during both training and inference phases. By dynamically managing memory through partitioning models into units and alternatingly transferring them between GPU and CPU, Superpipeline claims up to a 60% reduction in GPU memory consumption without compromising model accuracy. Unlike existing methods, which often focus on specific model types or use cases, Superpipeline is applicable to a wide range of models, including LLMs, VLMs, and vision models. The approach is adaptable and requires no model retraining, making it a versatile solution for hardware-constrained environments.

**Strengths:**

1. Broad Applicability: The method’s compatibility with diverse neural network architectures—beyond just LLMs—is a strong advantage, enhancing its usability across various AI fields.
2. Resource Efficiency: The reported GPU memory savings (up to 60%) are significant and enable larger model or batch processing on existing hardware.
3. Ease of Implementation: Superpipeline’s implementation does not necessitate complex model modifications or specific hardware requirements, allowing for straightforward integration.

**Weaknesses:**

1. Transfer Overhead: The performance impact due to GPU-CPU transfer time, particularly for specific models (e.g., ViT-bigG), limits the efficacy of Superpipeline in certain scenarios.
2. Scalability Concerns: The dependency on effective data transfer strategies (e.g., batch vs. sequential) could affect scalability, requiring additional customization for certain model or hardware setups.

**Questions:**

1. Can Superpipeline’s transfer limitations be mitigated for real-time applications? The delay in transfers could impact applications requiring fast inference cycles.

2. What are the specific requirements for tuning parameters k and k_prime for optimal performance? More details on optimal tuning strategies could enhance its usability for new users.

---

### Note · Authors · 2024-11-13

I have read and agree with the venue's withdrawal policy on behalf of myself and my co-authors.